# Single High-Dose Vitamin D Supplementation as an Approach for Reducing Ultramarathon-Induced Inflammation: A Double-Blind Randomized Controlled Trial

**DOI:** 10.3390/nu13041280

**Published:** 2021-04-13

**Authors:** Jan Mieszkowski, Andżelika Borkowska, Błażej Stankiewicz, Andrzej Kochanowicz, Bartłomiej Niespodziński, Marcin Surmiak, Tomasz Waldziński, Rafał Rola, Miroslav Petr, Jędrzej Antosiewicz

**Affiliations:** 1Department of Gymnastics and Dance, Gdańsk University of Physical Education and Sport, 80-336 Gdańsk, Poland; andrzejkochanowicz@o2.pl; 2Faculty of Physical Education and Sport, Charles University, 162 52 Prague, Czech Republic; petr@ftvs.cuni.cz; 3Department of Bioenergetics and Physiology of Exercise, Medical University of Gdańsk, 85-064 Gdańsk, Poland; andzelika.borkowska@gumed.edu.pl; 4Department of Biomedical Basis of Physical Education, Kazimierz Wielki University, Institute of Physical Education, 85-064 Bydgoszcz, Poland; blazej1975@interia.pl; 5Department of Anatomy and Biomechanics, Institute of Physical Education, Kazimierz Wielki University, 85-064 Bydgoszcz, Poland; bar.niespodzinski@wp.pl; 6Department of Internal Medicine, Jagiellonian University Medical College, 31-007 Krakow, Poland; marcin.surmiak@uj.edu.pl; 7Faculty of Health Sciences, Łomża State University of Applied Science, 18-400 Łomża, Poland; twaldzinski@pwsip.edu.pl; 8Masdiag Sp. Z O.O., 01-882 Warsaw, Poland; r.rola@doktorant.umk.pl; 9Chair of Environmental Chemistry and Bioanalytics, Faculty of Chemistry, Nicolaus Copernicus University, 87-100 Toruń, Poland

**Keywords:** IL-6, resistin, inflammation, ultramarathon, skeletal muscle damage, vitamin D

## Abstract

*Purpose*: A growing number of studies indicate the importance of vitamin D supplementation for sports performance. However, the effects of a single high-dose vitamin D supplementation on ultramarathon-induced inflammation have not been investigated. We here analyzed the effect of a single high-dose vitamin D supplementation on the inflammatory marker levels in ultramarathon runners after an ultramarathon run (maximal run 240 km). *Methods*: In the study, 35 runners (amateurs) were assigned into two groups: single high-dose vitamin D supplementation group, administered vitamin D (150,000 IU) in vegetable oil 24 h before the start of the run (*n* = 16); and placebo group (*n* = 19). Blood was collected for analysis 24 h before, immediately after, and 24 h after the run. *Results*: Serum 25(OH)D levels were significantly increased after the ultramarathon in both groups. The increase was greater in the vitamin D group than in the control group. Based on post-hoc and other analyses, the increase in interleukin 6 and 10, and resistin levels immediately after the run was significantly higher in runners in the control group than that in those in the supplementation group. Leptin, oncostatin M, and metalloproteinase tissue inhibitor levels were significantly decreased in both groups after the run, regardless of the supplementation. *Conclusions*: Ultramarathon significantly increases the serum 25(OH)D levels. Attenuation of changes in interleukin levels upon vitamin D supplementation confirmed that vitamin D has anti-inflammatory effect on exercise-induced inflammation.

## 1. Introduction

Excessive inflammatory response induced by exercise is thought to be one of the essential factors that limit sports performance. Proinflammatory cytokines can impair the metabolism of skeletal muscle and other tissues, e.g., insulin signaling. They can also induce inflammation of the central nervous system, which leads to the dysregulation of movement and coordination loss, in addition to general inflammation [1]. This might explain why many endurance athletes use non-steroid anti-inflammatory drugs even though they are ineffective or even toxic. For example, it has been demonstrated that plasma levels of some proinflammatory cytokines are increased in ultramarathon athletes who use ibuprofen [2], compared with non-users, and that ibuprofen use does not protect against the damage of skeletal muscle induced by eccentric exercise [3]. This shows the need for the identification of methods that would attenuate the increased inflammatory response among endurance athletes, such as ultramarathon runners.

Many studies suggest that vitamin D plays an anti-inflammatory role [4]. For example, it reduces the synthesis of tumor necrosis factor µ and interleukin (IL) 6 in monocytes [5]. Further, reduction of the inflammatory response by 1,25(OH)_2_D_3_, via upregulation of the inhibitor of nuclear factor κB, was observed in macrophages [6]. Vitamin D is synthetized in the skin. It is converted to 1,25(OH)_2_D_3_, its active form, via two hydroxylation reactions. Vitamin D shows endocrine effects as well as paracrine and autocrine functions. The endocrine effects are focused on controlling the calcium homeostasis in serum. The other functions are related to nuclear receptors of Vitamin D. [7]. Acting via vitamin D receptor (VDR), the active form of vitamin D regulates the expression of approximately 1000 genes, including those that encode proinflammatory cytokines. VDR is expressed in many tissues and cells, such as skeletal muscle, monocytes, and macrophages, and is critical for the modulation of inflammation.

Furthermore, vitamin D facilitates the anti-inflammatory effects of exercise training. For instance, inflammation induced by high-intensity exercise is significantly reduced upon vitamin D supplementation in an animal model [8]. This constitutes a good premise for the application of vitamin D supplementation in endurance runners to decrease the elevated inflammatory response; however, studies involving human subjects assessing the anti-inflammatory effects of vitamin D on inflammation induced by exercise are scarce. It was previously shown that single high dose of vitamin D can upregulate the vitamin D metabolites’ status in ultramarathon runners [9]. Accordingly, in the current study, we evaluated the impact of a single high-dose vitamin D supplementation on ultramarathon-induced inflammation.

## 2. Materials and Methods

### 2.1. Experimental Overview

This study is a continuation of previous investigation [9] aimed to assess the impact of single high dose of vitamin D on vitamin D metabolites in ultramarathon runners and is a part of the project: Vitamin D as a Factor Modifying Adaptation to Exercise.

This study was a double-blind randomized controlled trial with parallel groups, i.e., the supplementation and placebo (control) group. Supplementation involved administration of a single high dose of vitamin D.

During the initial visit (on 19–20 July 2018), information on the subjects’ age, body composition, and height was obtained. A professional physician examined all the runners. Venous blood was sampled 24 h before the ultramarathon (pre-supplementation), immediately after the run, and 24 h after the run. Vitamin D status and serum inflammatory marker levels were assessed at Gdańsk University of Physical Education (Gdańsk, Poland).

### 2.2. Participants

A group of 35 semi-professional ultramarathon runners (males) took part in the study. All participants started in the Lower Silesian Mountain Runs Festival 2018 Ultra Marathon Race. The participants completed an online questionnaire during the week before the run. They were randomly assigned to two groups: experimental (supplemented, S; *n* = 16) or placebo (control, C; *n* = 19) groups. Participants’ characteristics are summarized in Table 1 and the inclusion/exclusion criteria are presented in Box 1.

Box 1Eligibility criteria.
**Inclusion criteria**
1. Aged 30 years or older2. Experienced ultramarathon runner (minimum five starts in distances over 42 km)3. Healthy4. No additional drug intake or smoking and good health status5. No additional vitamin D and antioxidant supplementation
**Exclusion criteria**
1. Physically or mentally compromised individuals (currently treated for a psychiatric disorder, or alcohol or substance abuse), unwilling or unable to comply with study evaluations2. Comorbidities causing severe inflammation: Addison’s disease, allergy, asthma, celiac disease, psoriasis, Raynaud’s disease, rheumatoid arthritis, systemic lupus erythematosus, type 1 diabetes, and other.

All runners had previous ultramarathon experience (min. 5 starts). Enrolled participants completed survey aimed to define the methods and loads used during the training period (divided into a periods of general preparation and pre-start preparation) (Table 2). General preparation period lasted for 12 weeks prior to the pre-start preparation period before the start of the Lądek Zdrój Mountain Run. According to the survey, the pre-start preparation period lasted from 8 to 12 weeks. All ultramarathon runners confirmed the usage of pulse monitoring devices as an ongoing control tool in the training process (HR measurement). Additionally, in order to establish the approximate level of VO2max as a physical preparation state for the competition, subjects were asked to perform a Cooper test in a 12-min running version with the maximum possible intensity. The test was performed about 2 weeks before the start of the competition. Obtained results allowed to calculate (indirect method) the VO2 max indicator using the formula: VO2 max = (22.351 × distance covered in kilometers) − 11.288 [10] (Table 3). One week before the experiment and during all testing periods, the participants refrained from the intake of stimulants, such as alcohol, caffeine, chocolate, guarana, tea, or theine.

Similar eating patterns, based on a randomized diet for a corresponding age group and the intensity of physical activity, were devised for the participants, who were then asked to adopt them on the measurement days. The Bioethics Committee for Clinical Research at the Regional Medical Chamber in Gdańsk approved the study protocol (decision no. KB-24/16). The protocol was implemented in compliance with the Declaration of Helsinki. The subjects gave informed written consent before enrolling in the study. The study was registered as a clinical trial NCT03417700. Prior to participation, the subjects were informed about the study procedures; however, they were not aware of the rationale and study aim, and therefore were naive about the potential effects of supplementation. Power analysis for the interactions between effects, to determine the appropriate sample size, was performed in GPower ver. 3.1.9.2 [11]. The minimal total sample size for a medium effect size at the power of 0.8 and significance level of 0.05 was calculated as 28 subjects.

### 2.3. Ultramarathon Run

All participants took part in the Lower Silesian Mountain Run Festival 2018 organized at Lądek Zdrój (Lower Silesian Voivodeship, Poland). Race and race track characteristics: maximum course length: 240 km; maximum altitude: ca. 1425 m a.s.l.; minimum altitude: ca. 261 m a.s.l.; entire altitude range: ca. 1164 m; total ascent and descent: 7670 m; run start time: 18:00 h; temperature range during the run: from 18 °C (at the starting point) to 4 °C (on top of the Śnieżnik Mountain). No intensive rain or wind was registered during the run.

### 2.4. Vitamin D Supplementation

The athletes were randomly assigned to the S (supplemented) or C (control) groups. All participants in the S group were given a single high dose (150,000 IU) of vitamin D, as a solution in 10 mL of vegetable oil, 24 h before starting the ultramarathon. The placebo group received an equivalent volume of placebo solution. The taste (anise), consistency, and color of the placebo solution matched those of the vitamin D oil solution. No participants or researchers were aware of the group allocations, as the supplementation and placebo solutions were presented in carefully sealed sintered glass bottles marked with randomly assigned numbers.

### 2.5. Sample Collection, and Inflammation Marker and 25(OH)D Measurements

The blood was sampled by a medical diagnostic professional, according to the experimental protocol, i.e., at three times points: 24 h before the run, immediately after the run (within 5 min of the run finishing), and 24 h after the run.

The blood (9 mL) was collected into Sarstedt S-Monovette tubes (S-Monovette^®®^ Sarstedt AG&Co, Nümbrecht, Germany) containing a coagulation accelerator for serum separation. The serum was obtained by standard laboratory procedures, aliquoted into 500 µL portions, and frozen at −80 °C until analysis (at most 6 months).

For vitamin D analyses, serum proteins were first precipitated and derivatized. Quantitative analysis was done using liquid chromatography–tandem mass spectrometry [Shimadzu Nexera X2 UHPLC (Shimadzu, Japan)] coupled with 8050 triple quadruple detector (Shimadzu). The raw data were collected, processed, and quantified using LabSolutions LCGC. Concentration of the vitamin D metabolite 25(OH)D was determined at all sampling time points (before the start, immediately after, and 24 h after the end of the race).

Levels of the following inflammatory response and other serum markers were determined: follistatin-like 1 (FSTL-1), IL-6 and -15, leptin, leukemia inhibitory factor (LIF), oncostatin M (OSM), resistin, and tissue inhibitor of metalloproteinase 1 (TIMP-1). The analyses were done using a MAGPIX fluorescence detection system (Luminex Corp., Austin, TX, USA) with Luminex assays [Luminex Corp.; Luminex Human Magnetic Assay (13-Plex) LXSAHM-13].

### 2.6. Statistical Analysis

Descriptive statistics for all measured variables involved the mean ± standard deviation (SD). One-way ANOVA was performed to evaluate the differences in characteristics between groups. Two-way ANOVA with repeated measures (2 × 3) was used to determine the impact of ultramarathon (ultramarathon: 24 h before, immediately after, and 24 h after the run) on the 25(OH)D levels vs. vitamin D supplementation (group: S, C). Another set of two-way ANOVA analyses with repeated measures (group: S, C; ultramarathon: 24 h before, immediately after, and 24 h after the run) was used to evaluate the impact of ultramarathon running on the levels of inflammation markers vs. vitamin D supplementation. If significant interactions were detected, Tukey’s post-hoc test was used to determine the differences in specific subgroups. The assumption of normality and homogeneity of variances was checked by Shapiro-Wilk’s and Levene’s tests.

Pearson’s (r) correlation of changes in the serum 25(OH)D levels with changes in the inflammatory marker levels induced by ultramarathon was also analyzed. Eta-squared statistics (η2) were used to determine the effect size. Values equal to or exceeding 0.01, 0.06, and 0.14 indicated a small, moderate, and large effect, accordingly. All calculations were done, and graphics were generated, in Statistica 12 (StatSoft, Tulsa, OK, USA). Statistical significance was set at *p* ≤ 0.05.

## 3. Results

Table 1 contains a summary of the basic anthropometric characteristics of study participants. A significant difference between the groups was noted for height [F_(1, 33)_ = 13.81, *p* < 0.01, η2 = 0.34]. Other physical characteristics were not statistically different between the groups.

The performance of ultramarathon runners are shown in Table 4.

Table 5 summarizes changes in the vitamin D serum levels immediately after the ultramarathon. There were no differences in serum levels of vitamin D between groups before the ultramarathon Two-way ANOVA with repeated measures indicated a significant effect of the *group* factor [F_(1,33)_ = 6.60, *p* < 0.05, η2 = 0.19] and *ultramarathon* [F_(1, 33)_ = 67.26, *p* < 0.01, η2 = 0.70]. Significant interactions between the *group* factor and *ultramarathon* were noted [F_(2,66)_ = 7.39, *p* < 0.01, η2 = 0.21]. Post-hoc analysis confirmed the increase of 25(OH)D levels immediately after the run in both groups, and also indicated that of the two groups, the increase was more pronounced in group S (Table 5).

Figure 1 shows the effect of single high-dose vitamin D supplementation on the inflammatory marker levels induced by the ultramarathon. There were no differences in measured biomechanical markers between groups before the ultramarathon. Figure 2 presents the changes in the selected biochemical markers induced by the ultramarathon run. Table 6 presents the analysis (two-way ANOVA) of changes in the selected biochemical markers induced by the ultramarathon. A significant effect of the ultramarathon was detected on all analyzed biochemical markers except for LIF. Regardless of the vitamin D supplementation, a significant increase in the FSTL-1 (20.2%), IL-6 (391.4%), IL-10 (23.7%), and resistin (78.2%) levels, and a significant decrease in the leptin (27.1%), OSM (5.0%), and TIMP-1 (29.9%) levels were observed immediately after the run. ANOVA of the IL-6, -10, and -15, resistin, and TIMP-1 levels indicated a significant interaction of the *group* and *ultramarathon* (Table 6). As determined by post-hoc and other analyses, the increase in the IL-6 and -10, and resistin levels immediately after the ultramarathon in runners in group C was significantly higher than that in runners in group S. All values measured 24 h after the run were comparable with the baseline values, except for the IL-15 levels, which were significantly reduced 24 h after the run in group S.

Based on Pearson’s (r) correlation analysis of changes in the serum vitamin D levels with changes in the inflammatory marker levels induced by the ultramarathon, changes in the 25(OH)D and IL-6 levels immediately after the ultramarathon were significantly negatively correlated in group S (Table 7). Further, changes in the vitamin D levels immediately after and 24 h after the ultramarathon were also significantly negatively correlated with changes in the IL-15 levels 24 h after the run. On the other hand, in group C, changes in the vitamin D levels 24 h after the run were negatively correlated with those in IL-15 levels immediately after the run. Finally, in group C, changes in the vitamin D levels immediately after the run and 24 h after the run were significantly positively correlated with those in leptin immediately after the run.

## 4. Discussion

We here demonstrated that the inflammatory response induced by ultramarathon running is significantly blunted in runners who received a single high dose of vitamin D before the run. This novel outcome is important from the perspective of post-exercise regeneration. According to some reports, a portion of vitamin D is liberated into the blood during exercise [12]. Here, accordingly to our previous study [9], we observed that the levels of 25(OH)D, a vitamin D status marker, significantly increased after the ultramarathon, more so in the supplemented group than in the control group. Hence, elevated vitamin D levels play a key role in the modulation of the inflammatory response in terms of serum marker levels. The outcome varied for the individual cytokines.

Studies on the anti-inflammatory effect of vitamin D on exercise-induced inflammation involving human subjects are limited and inconclusive. Further, in many disease states, the link between vitamin D status and inflammation is ambiguous. For example, an inverse association between the levels of 25(OH)D_3_ and tumor necrosis factor α was reported for endurance-trained runners [13]. By contrast, no significant changes in the levels of tumor necrosis factor α and IL-6 were reported in healthy overweight and obese subjects supplemented with a high dose of vitamin D after 12 weeks of progressive resistance exercise training [14]. Hence, if appropriate vitamin D levels are ensured, they can be expected to have an anti-inflammatory effect, with the effect of additional vitamin D supplementation less pronounced than in the case of vitamin D insufficiency. The observed significant increase in the levels of IL-6 and -10, and resistin after the ultramarathon was attenuated in runners supplemented with vitamin D. Conversely, vitamin D supplementation had no effect on the levels of another proinflammatory cytokine, FSTL1, which significantly increased after the run. These observations indicate that vitamin D attenuates exercise-induced inflammation.

### 4.1. IL-6

IL-6 plays a metabolic role and also has a regulatory role in inflammation. While IL-6 levels increase during exercise stimulates fatty acid mobilization, it also stimulates the inflammatory response. Inhibition of IL-6 signaling ameliorates inflammatory diseases [15]. In one study involving morbidly obese subjects, 25(OH)D levels showed a significant inverse correlation with resting IL-6 levels [16]. We made similar observations herein, considering changes in the IL-6 levels immediately after the ultramarathon in the vitamin D-supplemented group. Reduced IL-6 levels were also observed 24 h after eccentric exercise in athletes supplemented over the course of 3 weeks with a low dose of vitamin D (2 × 1000 IU/day) [17]. In another study, involving elderly men, an inverse correlation was reported between the VDR and intramuscular IL-6 levels. Furthermore, vitamin D stimulates VDR expression [18]. It was shown that VDR levels were inversely correlated with the levels of intramuscular IL-6. Taken together, the observations made in the current study support the notion that vitamin D status determines the serum IL-6 levels. To the best of our knowledge, this is the first time that attenuation of IL-6 level increase after exercise by administration of a single high dose of vitamin D was shown in human.

### 4.2. IL-15

It has been reported that both, serum IL-15 and skeletal muscle IL-15 mRNA levels do not change after 2.5–3 h of running [19,20]. While the effort applied in the current study was longer lasting, the outcome in C group was similar to results reported in mentioned studies. Interestingly, a decrease in IL-15 concentration 24 h after the ultramarathon was observed in group S. One explanation for this outcome involves Toll-like receptor 2 (TLR2). TLR-2–dependent activation of IL-15 gene expression was previously shown [21]. Furthermore, supplementation with vitamin D decreases TLR2 levels in human monocytes, in a dose- and time-dependent manner, with the maximum effect after 72 h [22]. Collectively, this may indicate that increasing the concentration of vitamin D 24 h after the run might play a role in the reduction of IL-15 levels; however, more detailed studies are needed to confirm the involvement of TLR2.

### 4.3. Resistin

Resistin is a proinflammatory cytokine secreted by various tissues. In humans, cells of the immune system, i.e., lymphocytes, monocytes, and macrophages, are its main secreting cells [23,24,25]. After its discovery in 2001, its function has been mainly associated with obesity, type 2 diabetes, and broadly understood insulin resistance, hence the name [26]. In addition to modulating insulin sensitivity, resistin plays a role in the development of inflammation and inflammatory diseases (e.g., atherosclerosis and arthritis), regulates carbohydrate and lipid metabolism, and stimulates the proliferation of endothelial cells [27]. Of note, regular physical activity significantly reduces the levels of resistin [28] and high-density lipoprotein cholesterol, thus preventing inflammation and lowering the risk of heart disease [29]. Here, we showed that serum resistin levels increase significantly in ultramarathoners immediately after the run. These observations confirm an earlier report from a Greek group on a significant increase of this parameter after an ultramarathon run [30]. We also showed here that a single high-dose supplementation with vitamin D inhibits the increase of exercise-induced resistin levels in runners. The effect of vitamin D on resistin levels in athletes has not been studied until now. Further, the relationship between resistin and vitamin D has only been examined in a few studies. However, these studies mainly involved obese subjects and related diseases, as well as diseases involving chronic inflammation [31,32]. Increased levels of resistin in the muscle cell during exercise appear to be the factor limiting exercise capacity, and the observation that vitamin D supplementation lowers resistin levels is notable. Additional studies are needed to understand the role that resistin plays in exercise and inflammation.

### 4.4. IL-10

IL-10 is an anti-inflammatory cytokine whose blood levels increase after exercise. We confirmed this in the current study, as we observed that IL-10 levels significantly increased after the run. Intriguingly, this effect was not apparent in runners supplemented with vitamin D. Consistently with Matilainen et al. [33], the *IL10* gene contains regions that recruit VDR in a ligand-dependent fashion and 1,25(OH)D_3_ reduces *IL10* expression. Further, the release of IL-6 from the exercising muscle is accompanied by IL-10 and other anti-inflammatory cytokine level increase in the circulation [34]. Hence, IL-10 level reduction after a run can be also explained by a reduced stimulation of IL-10 synthesis by IL-6.

Finally, leptin, OSM, and LIF have been implicated in inflammation [35,36,37]. Contrary to our expectations, leptin and OSM levels significantly decreased after the run, and vitamin D did not affect these changes.

In conclusion, we showed here that administration of a single high dose of vitamin D significantly blunts the rise of proinflammatory cytokine levels after an ultramarathon, even though serum levels of 25(OH)D are significantly elevated after the run. These observations imply that the ultramarathon-induced increase in 25(OH)D levels is not sufficiently high so as to reduce inflammation. Hence, improving the vitamin D status before an endurance competition might be a good alternative to the use of anti-inflammatory drugs that are so often relied on in sports.

## Figures and Tables

**Figure 1 nutrients-13-01280-f001:**
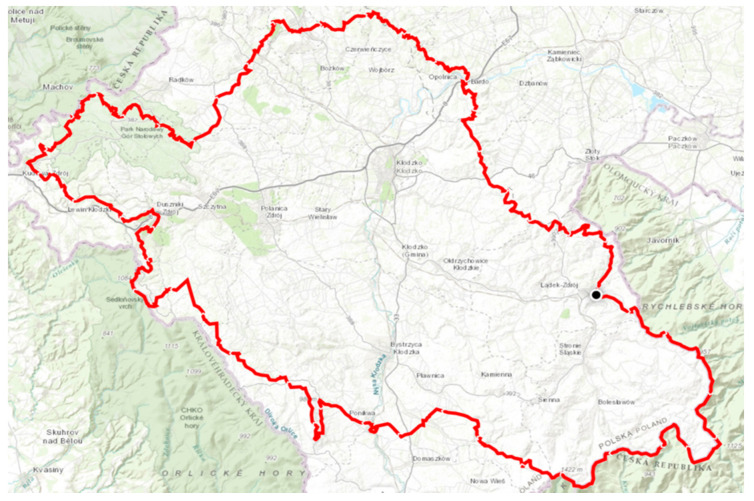
Ultramarathon track characteristics (Lower Silesian Mountain Run Festival 2018, Lądek Zdrój) (Mountain Marathons Foundation).

**Figure 2 nutrients-13-01280-f002:**
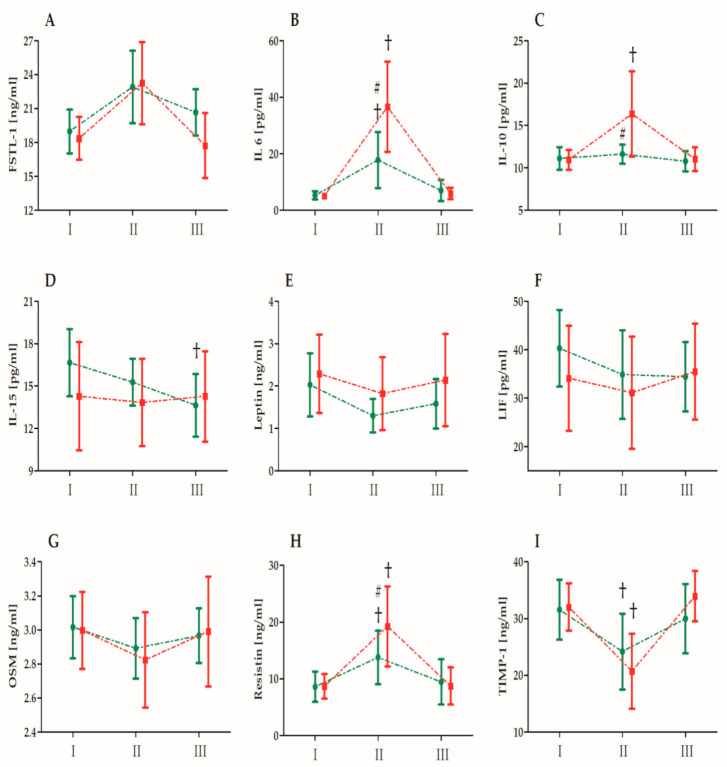
Changes in biochemical marker levels after the ultramarathon in runners who receiveda single high dose of vitamin D (group S, green) and runners who received the placebo (group C, red). Sampling I, 24 h before the run; II, immediately after the run; and III, 24 h after the run. Abbreviations: FSTL-1, follistatin-like 1; IL, interleukin; LIF, leukemia inhibitory factor; OSM, oncostatin M; and TIMP-1, tissue inhibitor of metalloproteinase 1. †, Significant difference vs. 24 h before the run; #, significant difference vs. group C immediately after the run. The significance level was set at *p* < 0.01.

**Table 1 nutrients-13-01280-t001:** Participants’ characteristics (*n* = 35).

_Variable_	_Supplemented Group_ _(*N =* 16)_	_Control Group_ _(*N =* 19)_	*_p_*	Effect Size (η^2^)
	_Mean ± SD_	_Mean ± SD_		
_Age (years)_	_42.40 ± 7.59_	_39.48 ± 6.89_	_0.21_	_0.04_
_Body height (cm)_	_175.20 ± 4.34 *_	_179.67 ± 4.64_	_0.01_	_0.17_
_Body mass_	_72.51 ± 6.71_	_76.19 ± 5.25_	_0.07_	_0.08_
_Body mass index (kg/m_ ^2^ _)_	_23.24 ± 2.78_	_24.45 ± 1.19_	_0.11_	_0.06_
_Fat mass (%)_	_12.13 ± 3.89_	_12.85 ± 4.42_	_0.36_	_0.03_

**Note:** *, significant difference vs. control group at *p* < 0.05.

**Table 2 nutrients-13-01280-t002:** Summary of training loads during typical one week of the two periods of training (*N* = 35).

		Number of Training Units Per Week	CR 1 (Km)	CR 2 (Km)	CROSS 1 (Km)	CROSS 2 (Km)	Speed (Km)
General preparation period	Mean	5.00	60.94	11.58	7.58	3.91	0.88
	SD	0.83	16.29	4.21	2.78	2.88	0.60
Pre-start preparation period	Mean	5.70	67.39	13.52	14.38	5.7	1.57
	SD	0.85	11.96	2.60	4.35	2.65	0.59

**Note:** CR 1–70–80% HR max—continuous running in the first intensity range (70–80% HRmax); CR 2–80–90% HR max—continuous running in the second intensity range (80–90% HRmax); CROSS 1—up-downhill running in different tempo (75–85% HR max); CROSS 2–up-downhill running in different tempo (85–95% HR max); Speed–100–200 meters distance running with high intensity.

**Table 3 nutrients-13-01280-t003:** Characteristics of the maximum oxygen uptake capacity (VO_2_ max) in ultramarathon runners basing on the Cooper test (Cooper, 1968).

Variable	Supplemented Group (*N* = 16)	Control Group (*N* = 19)	P	Effect Size (η^2^)
	Mean ± SD	Mean ± SD		
VO_2_max (mL × kg^−1^ × min^−1^)	53.73 ± 6.04	54.40 ± 5.68	0.74	<0.01
Distance (km)	2.908 ± 0.263	2.939 ± 0.254	0.75	<0.01

**Note:** VO_2max_—maximal oxygen uptake.

**Table 4 nutrients-13-01280-t004:** Ultramarathon runners performance at particular distances.

Distance	Supplemented Group (*N* = 16)	Control Group (*N* = 19)	*p*	Effect Size (η^2^)
	Mean ± SD (hh:mm:ss) ± (hh:mm:ss)	Mean ± SD (hh:mm:ss) ± (hh:mm:ss)		
10 km	01:18:49 ± 00:09:28	01:19:28 ± 00:10:32	0.74	<0.01
32 km	05:02:36 ± 01:06:45	04:44:09 ± 00:59:51	0.32	0.03
64 km	10:15:03 ± 01:30:26	09:47:07 ± 00:59:51	0.27	0.04
100 km	15:57:07 ± 01:52:45	15:45:43 ± 01:42:34	0.78	<0.01
130 km	21:07:20 ± 02:12:27	19:01:51 ± 02:17:26	0.12	0.06
170 km	29:40:07 ± 02:55:52	27:57:05 ± 02:23:56	0.16	0.05
215 km	39:47:07 ± 04:01:08	37:35:43 ± 05:00:51	0.23	0.04
240 km	44:46:33 ± 04:59:23	42:01:35 ± 05:40:04	0.25	0.04

**Table 5 nutrients-13-01280-t005:** Two-way ANOVA (2 groups × 3 repeated measurements) of changes in the serum vitamin D levels induced by the ultramarathon.

Variable	Group	24 H Before The Run (Mean ± SD)	Immediately after The Run (Mean ± SD)	24 H after The Run (Mean ± SD)
25(OH)D [ng/mL]	Supplemented Control	27.50 ± 7.01 26.82 ± 5.22	58.13 ± 18.89 †# 41.06 ± 10.58 †	67.93 ± 25.67 † 49.54 ± 17.76 †

**Note**: Post-hoc analysis: †, significant difference vs. 24 h before the run; #, significant difference vs. group C immediately after the run. The significance level was set at *p* < 0.01.

**Table 6 nutrients-13-01280-t006:** Two-way ANOVA (2 groups × 3 repeated measures) of changes in the biochemical marker levels induced by the ultramarathon.

Variable	Effect	F	Df	P	Effect Size (η^2^)	Post-Hoc Outcome
FSTL-1	GR UM GR × UM	3.26 18.29 1.76	1, 33 2, 66 2, 66	0.08 0.01 ** 0.18	0.09 0.35 0.05	I, III < II
IL-6	GR UM GR × UM	5.03 55.84 8.35	1, 33 2, 66 2, 66	0.03 * 0.01 ** 0.01 **	0.19 0.70 0.20	S < C I, III < II S-I, S-III < S-II C-I, C-III < C-II S-II < C-II
IL-10	GR UM GR × UM	6.25 13.58 7.37	1, 33 2, 66 2, 66	0.02 * 0.01 ** 0.01 **	0.08 0.62 0.11	S < C I, III < II C-I, C-III < C-II S-II < C-II
IL-15	GR UM GR × UM	2.17 3.52 3.61	1, 33 2, 66 2, 66	0.14 0.03 * 0.03 *	0.06 0.10 0.10	I > III S-I > S-III
Leptin	GR UM GR × UM	3.00 8.65 0.63	1, 33 2, 66 2, 66	0.09 0.01 ** 0.53	0.08 0.21 0.02	I > II
LIF	GR UM GR × UM	2.11 1.79 1.36	1, 33 2, 66 2, 66	0.15 0.17 0.26	0.06 0.05 0.04	
OSM	GR UM GR × UM	0.11 4.06 0.34	1, 33 2, 66 2, 66	0.73 0.02* 0.70	0.01 0.11 0.01	I > II
Resistin	GR UM GR × UM	1.20 36.6 4.20	1, 33 2, 66 2, 66	0.27 0.01 ** 0.02*	0.03 0.52 0.11	I, III < II S-I, S-III < S-II C-I, C-III < C-II S-II < C-II
TIMP-1	GR UM GR × UM	0.07 35.98 4.19	1, 33 2, 66 2, 66	0.78 0.01 ** 0.02 *	0.01 0.52 0.11	I, III > II S-I, S-III > S-II C-I, C-III > C-II

**Note:** Markers: FSTL-1, follistatin-like 1; IL, interleukin; LIF, leukemia inhibitory factor; OSM, oncostatin M; TIMP-1, tissue inhibitor of metalloproteinase 1. Study design: GR, group; S, runners who received a single high dose of vitamin D; C, runners who received the placebo (control group); UM, ultramarathon; I, 24 h before the run; II, immediately after the run; and III, 24 h after the run. Significant difference detected at * *p* ≤ 0.05 or ** *p* ≤ 0.01.

**Table 7 nutrients-13-01280-t007:** Analysis of Pearson’s correlation (r) of changes in the serum vitamin D levels with ultramarathon-induced changes in the inflammatory marker levels.

_Variable_	_Change_	_Supplemented Group_ _(*n* = 16)_		_Control Group_ _(*n* = 19)_	
		_Δ I–II_	_Δ I–III_	_Δ I–II_	_Δ I–III_
_FSTL-1_	_Δ I–II_	_0.02_	_0.03_	_0.18_	_0.09_
	_Δ I–III_	_0.31_	_0.24_	_0.19_	_0.12_
_IL-6_	_Δ I–II_	_−0.37 *_	_−0.34_	_−0.33_	_−0.26_
	_Δ I–III_	_0.03_	_−0.01_	_−0.12_	_−0.12_
_IL-10_	_Δ I–II_	_−0.26_	_−0.19_	_−0.14_	_−0.11_
	_Δ I–III_	_0.14_	_0.13_	_0.06_	_−0.08_
_IL-15_	_Δ I–II_	_−0.27_	_−0.33_	_−0.40_	_−0.50 *_
	_Δ I–III_	_−0.41 *_	_−0.45 *_	_−0.13_	_−0.28_
_Leptin_	_Δ I–II_	_0.10_	_0.20_	_0.50 *_	_0.58 *_
	_Δ I–III_	_−0.11_	_−0.15_	_−0.27_	_−0.31_
_LIF_	_Δ I–II_	_0.14_	_0.20_	_0.37_	_0.47_
	_Δ I–III_	_−0.19_	_−0.18_	_−0.06_	_−0.13_
_Resistin_	_Δ I–II_	_−0.12_	_−0.08_	_0.14_	_0.08_
	_Δ I–III_	_0.17_	_0.21_	_0.26_	_0.33_
_OSM_	_Δ I–II_	_0.03_	_−0.14_	_−0.22_	_−0.43_
	_Δ I–III_	_0.07_	_0.03_	_0.40_	_0.29_
_TIMP-1_	_Δ I–II_ _Δ I–III_	_0.18_ _−0.23_	_0.12_ _−0.22_	_−0.07_ _−0.01_	_−0.09_ _−0.10_

**Note**: Markers: FSTL-1, follistatin-like 1; IL, interleukin; LIF, leukemia inhibitory factor; OSM, oncostatin M; TIMP-1, tissue inhibitor of metalloproteinase 1. Study design: Δ I–II, difference between the values before and immediately after the ultramarathon; Δ I–III, difference between the values before and 24 h after the ultramarathon. Significant difference detected at * *p* ≤ 0.05.

## Data Availability

The data that support the findings of this study are available on request from the corresponding authors J.M. and J.A.

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
