# Peer review of "Single High-Dose Vitamin D Supplementation as an Approach for Reducing Ultramarathon-Induced Inflammation: A Double-Blind Randomized Controlled Trial"

_nutrients, 2021, doi:10.3390/nu13041280_

Round 1
Reviewer 1 Report
An interesting study on the use of high doses of vitamin D in marathon runners; I have some queries:
You did not mention smoking in the exclusion criteria...I think it should be added.
No need to specify from what vein blood was collected ( I also find it very unlikely that in 35 patients all blood samples were from a single vein).
A small description of vitamin D and its function would be a great addition to the introduction; here's an article you should consider: doi: 10.1007/s13668-020-00322-4.
Thank You
Author Response
Authors would like to thank the editor and reviewers for the opportunity to revise and for all comments that will improve the quality of the manuscript. Please find below direct response to the comments.
Please notice that all line indication refer to the version with highlighted changes. Aside the comments below, to clarify all the methods, we introduced the extended information about procedures and we erased some results in the text not to copies those conclusions that were already mentioned in previsions works. Please note that text of the manuscript was extensively modified to meet the Journal’s quality standards as requested by the Editor.
An interesting study on the use of high doses of vitamin D in marathon runners; I have some queries: No need to specify from what vein blood was collected ( I also find it very unlikely that in 35 patients all blood samples were from a single vein).
A small description of vitamin D and its function would be a great addition to the introduction; here's an article you should consider: doi: 10.1007/s13668-020-00322-4. Thank You
You did not mention smoking in the exclusion criteria...I think it should be added.
Answer: thank you for this point. Considering the study design, mentioned smoking was one of the criteria, but we did not write about it – it was a small editorial error, w already corrected it and now it is in the table [Line 109]. None of the runners mark smoking but it was one of the health related behaviours marked in the preliminary survey, before the experiment.
No need to specify from what vein blood was collected ( I also find it very unlikely that in 35 patients all blood samples were from a single vein).
Answer: thank you for this point. Blood samples were drawn by a qualified medical diagnostic professional from the venous in the area of the elbow flexion and actually concerned not only one vessel but available at a given stage of the study. At any time of the study samples were drawn by same medical diagnostic professional. This information could be misleading and it has been changed accordingly [Line 156-160].
A small description of vitamin D and its function would be a great addition to the introduction; here's an article you should consider: doi: 10.1007/s13668-020-00322-4.
Answer: thank you for this comment. The information about vitamin D and its function in a human body, specially from suggested article has been added to the text [Line 54-56, 70-73]
Reviewer 2 Report
The authors present an interesting study where they try to analysis the effect of supplementing with vitamin D to reduce inflammation.
Firstly, I can see that authors have published a manuscript very similar in nutrients the last year. I do not understand the differences in the number of samples 35 vs 27. In addition, it is repeated results and conclusion already published in the previous article such as an increase in the vitamin D levels. So, authors must reference these results and not repeat in the present work.
Introduction
Authors talk about vitamin D regulate around 900 genes, but in the previous manuscript mentioned an action over 1000 genes. It must be a harmony among both articles.
Methods
Although authors describe that participants are semi-professional who have ran at least 5 competitions. It is important to know their fitness levels and how long ago they ran such tests, given it has been observed the fitness level affect a post-test state and might influence on biomarkers levels.
It is not show if it has been analysed the normality.
Authors focus their comparation on after ultramarathon moment. However, they did not show whether there is differences between groups (S,C) at basal stage or at the following day.
Author Response
The authors would like to thank the editor and reviewers for the opportunity to revise and for all comments that will improve the quality of the manuscript.
Please find below a direct response to the comments.
Please notice that all line indications refer to the version with highlighted changes. Aside from the comments below, to clarify all the methods, we introduced the extended information about procedures and we erased some results in the text not to copies those conclusions that were already mentioned in previsions works. Please note that text of the manuscript was extensively modified to meet the Journal’s quality standards as requested by the Editor.
Reviewer
Comments and Suggestions for Authors
The authors present an interesting study where they try to analysis the effect of supplementing with vitamin D to reduce inflammation.
Firstly, I can see that authors have published a manuscript very similar in nutrients the last year. I do not understand the differences in the number of samples 35 vs 27.
Answer: thank you for this comment. Here we present data on 25(OH)D however in the previous study all the metabolites of vitamin D were measured for the limited number of participants. In the present study 25(OH)D and cytokines were measured in all of them. This situation is caused by the statistical significance of presented data and the cost of all analyzes.
In addition, it is repeated results and conclusion already published in the previous article such as an increase in the vitamin D levels. So, authors must reference these results and not repeat in the present work.
Answer: thank you for this comment. Such a situation shouldn’t take place. It was only mentioned in the text to show the real supplementary effect associated with vitamin D supplementation. We delated when it wasn’t needed and we added citations in the text where the article should be referenced.
Introduction
Authors talk about vitamin D regulate around 900 genes, but in the previous manuscript mentioned an action over 1000 genes. It must be a harmony among both articles.
Answer: thank you for this comment. Reviewer is right vitamin D regulates action over 1000 genes so we corrected this editorial error. Now there is harmony among both articles [Line 67].
Methods
Although authors describe that participants are semi-professional who have ran at least 5 competitions. It is important to know their fitness levels and how long ago they ran such tests, given it has been observed the fitness level affect a post-test state and might influence on biomarkers levels.
Answer: thank you for a good point. Indeed, each of the runners has already had running experience related to the start in an ultramarathon, but in a given starting year it was the first run of such a long-lasting ultra running. Previsions run for every one of the population was not less than 6 months from the start of the study. We are aware that fitness levels can contribute to different responses after the exercise. Yet, our study aim was focused on the effects of a single high dose of vitamin D on ultramarathon-induced inflammation in general, thus a variety of fitness levels was necessary to show if the impact of vitamin D could be generalized. The question of how the different fitness levels can modulate this general effect is another scientific problem, which goes beyond this elaboration but we are encouraged to consider it in future investigations.
It is not show if it has been analysed the normality.
Answer: thank you for this comment. The information was added: The assumption of normality and homogeneity of variances was checked by Shapiro-Wilk’s and Levene’s tests [Line 190-195].
Authors focus their comparation on after ultramarathon moment. However, they did not show whether there is differences between groups (S,C) at basal stage or at the following day.
Answer: Thank you for this comment. We believe that the information about baseline (before) ultramarathon is shown. Please look at Figure 2. as well as corresponding Table 3. Each “I” indicate status before the ultramarathon. None of the investigated markers showed differences at baseline between the two groups, as presented by the post hoc test for the ANOVA. Perhaps, despite the description in methods it was not clearly shown in results sections. We made changes accordingly.

Round 2
Reviewer 1 Report
The authors responded to all queries. The article is ready to be published.
Author Response
Authors would like to thank the reviewer for all comments that helped to will improve the quality of the manuscript.
Reviewer 2 Report
The authors answer almost all the questions raised.
Author Response
Authors would like to thank the reviewer for all comments that helped to improve the quality of the manuscript. According to the review we added few new informations in the text about the about the performance and physical fitness of the runners. We think that it helped to clarify all doubts related to competition and intergroup differences.